# Mitigating Spatial Redundancy: A Predictive Compression Framework for 3D Gaussian Splatting

## Abstract

3D Gaussian Splatting (3DGS) has emerged as a promising framework for Novel View Synthesis (NVS) due to its superior rendering quality and real-time performance. However, its widespread adoption is hindered by the substantial storage and transmission costs associated with the massive number of primitives. Notably, existing 3DGS compression approaches encode every primitive in its entirety, failing to utilize spatial continuity to compress shared content across primitives. In this work, we propose **Predict-GS**, a predictive compression framework for anchor-based Gaussian to mitigate spatial redundancy among anchors. Specifically, we construct a Spatial Feature Pool (SFP) based on a hybrid representation of multi-resolution 3D grids and 2D planes, which serves to predict coarse Gaussians for scene reconstruction. To refine these predictions, we introduce a residual compensation module equipped with a Multi-head Gaussian Residual Decoder (MGRD) that models corrections for shape and appearance, thereby transforming coarse Gaussians into high-fidelity ones. Furthermore, we revisit the inherent characteristics of our framework and design a prediction-tailored progressive training strategy to enhance its effectiveness. Extensive experiments on public benchmarks demonstrate the effectiveness of our framework, achieving a remarkable size reduction of over 58× compared to vanilla 3DGS on Mip-NeRF360 and outperforming the state-of-the-art (SOTA) compression method.

## 1 Introduction

Novel View Synthesis (NVS) plays a pivotal role in Augmented Reality (AR) Franke et al. (2024), Virtual Reality (VR) Zhai et al. (2025) and Mixed Reality (MR) Speicher et al. (2019). Approaches based on implicit Neural Radiance Fields (NeRF) Mildenhall et al. (2021) have demonstrated remarkable capabilities in generating photorealistic novel views by accumulating RGB values along sampled rays. However, NeRF-based methods Mildenhall et al. (2021); Tosi et al. (2023) heavily rely on repeated queries to a Multi-Layer Perceptron (MLP), resulting in substantial computational overhead and prohibitive training and rendering time. Recently, 3D Gaussian Splatting (3DGS) Kerbl et al. (2023) has emerged as a compelling alternative, which models scenes using anisotropic 3D Gaussian primitives with learnable attributes. Due to the high fidelity and real-time performance, 3DGS has inspired a series of works Huang et al. (2024); Guédon & Lepetit (2024); Zhang et al. (2024); Ye et al. (2024); Kheradmand et al. (2025); Wu et al. (2024); Yang et al. (2024b); Liang et al. (2024); Tang et al. (2023); Wu et al. (2025); Yan et al. (2024); Rota Bulò et al. (2024); Lu et al. (2024a); Yu et al. (2024b;a); Yang et al. (2024a) in NVS community and is widely considered a promising representation for 3D reconstruction.

However, 3DGS requires storing a large number of 3D Gaussian primitives along with their complex attributes, resulting in substantial storage overhead—often reaching several hundred megabytes or even exceeding 1 gigabyte per scene Bagdasarian et al. (2024). The prohibitive storage and transmission costs pose challenges to the deployment in scenarios involving resource-constrained devices or real-time streaming requirements. Existing 3DGS compression methods can be broadly classified into four primary strategies: **(a)** Designing compact representation Lu et al. (2024b); Ren et al. (2024), **(b)** Pruning redundant Gaussian primitives Mallick et al. (2024); Fang & Wang (2024a;b); Lee et al. (2024), **(c)** Adaptive quantization Lee et al. (2024); Fan et al. (2024), and

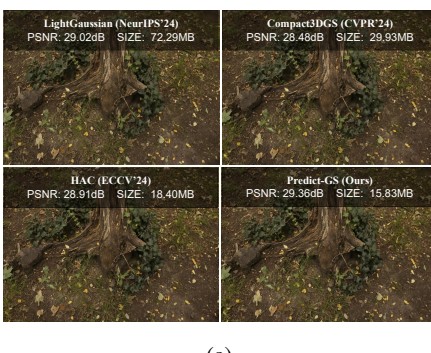 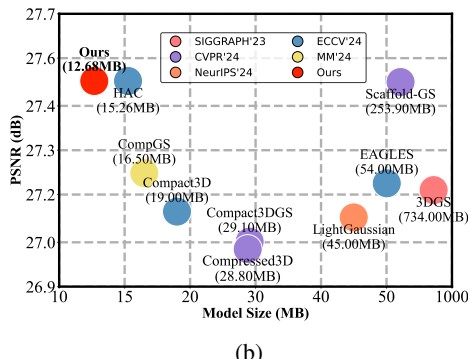

(a)                                            (b)

Figure 1: **(a)** presents an intuitive comparison between our method and representative 3DGS compression methods, LightGaussian Fan et al. (2024), Compact3DGS Lee et al. (2024) and HAC Chen et al. (2024b) on *stump* scene in Mip-NeRF360 Barron et al. (2022) dataset. **(b)** compares the performance of representative methods in the field of 3DGS compression. Methods positioned closer to the top-left corner indicate better compression performance. The results are evaluated on Mip-NeRF360 Barron et al. (2022) dataset. Our method compresses vanilla 3DGS by 58× and even achieves a 16.91% bit rate reduction compared to the state-of-the-art.

**(d)** Eliminating statistical redundancy across Gaussian attributes, typically through learned entropy models Chen et al. (2024b); Wang et al. (2024); Liu et al. (2024a); Chen et al. (2025) for entropy coding. Despite these advances, the compression performance requires further improvement to support real-time transmission and deployment on storage-constrained devices.

Existing 3DGS compression methods Lee et al. (2024); Niedermayr et al. (2024); Fan et al. (2024); Girish et al. (2024); Liu et al. (2024c); Navaneet et al. (2024); Chen et al. (2024b; 2025); Wang et al. (2024); Liu et al. (2024a) are fundamentally grounded in a post-compression paradigm, where techniques such as quantization and entropy coding are applied after scene reconstruction. A notable limitation is that these post-compression methods have to encode each primitive as a whole, without exploiting the inherent spatial redundancy across them, since each primitive is treated as an independent entity during the reconstruction stage Kerbl et al. (2023); Lu et al. (2024b). In practice, however, 3D scenes exhibit continuous spatial structures, where nearby primitives often possess highly correlated attributes. This implies that, in post-compression methods, shared features across primitives are redundantly stored and encoded. While Scaffold-GS Lu et al. (2024b) proposes using anchors to aggregate Gaussians and thereby reduce spatial redundancy among Gaussians within a local neighborhood, it is important to note that the aggregation remains fine-grained and does not capture large-scale context. As a result, **spatial redundancy at anchor level still persists**.

Prediction techniques have been employed in image compression Minnen et al. (2018); Ma et al. (2019) to effectively reduce spatial redundancy among neighboring pixels. Specifically, rather than encoding every pixel individually, only some pixels are explicitly encoded while the others are predicted from the former. However, predictive strategies remain unexplored in 3DGS compression due to the following challenges: **(a)** The spatial structure of Gaussians or anchors is sparse and irregular, lacking the well-defined neighborhood patterns found in images that are essential for prediction-based techniques. **(b)** Forcibly constructing such neighborhoods introduces significant computational overhead. **(c)** Gaussians or anchors move continuously during the training process, which causes dynamic neighborhood changes, further complicating the prediction process. For the reasons above, it is challenging to build a prediction-driven compression framework for 3DGS.

In this work, we propose a predictive compression framework (**Predict-GS**) for anchor-based Gaussian Lu et al. (2024b), as it offers a more compact structure and serves as the foundation for many representative 3DGS compression methods Chen et al. (2024b); Ren et al. (2024); Wang et al. (2024); Ma et al. (2025); Chen et al. (2025). Specifically, we construct a lightweight Spatial Feature Pool (SFP) to compactly store shared features for representing the 3D scene, enabling each anchor to query the spatial feature at its location for direct Gaussian primitive prediction. However, the predicted Gaussians struggle to accurately capture fine-grained details and high-frequency textures in the scene. To address this issue, we introduce a residual compensation strategy, which adjusts the

predicted Gaussians' shape and appearance by adding correction terms. By integrating Gaussian prediction with residual compensation, we achieve spatial redundancy elimination for anchor feature, i.e., instead of encoding the complete feature for every anchor, we only retain their residual. Moreover, we observe that the predictive compression framework suffers from suboptimal reconstruction quality under the conventional training pipeline, primarily due to a mismatch between the limited capacity of the SFP and the complexity of the training views. To address this discrepancy, we design a prediction-tailored progressive training strategy to alleviate optimization difficulties and enhance the performance of our framework. In all, our contributions are summarized as follows:

- We propose a prediction-driven 3DGS compression framework (**Predict-GS**) that leverages a Spatial Feature Pool (SFP) for Gaussian prediction and a residual compensation module for refinement, effectively reducing spatial redundancy while maintaining high rendering quality.

- To fully exploit the potential of our predictive compression framework, we design a prediction-tailored progressive training scheme for improving optimization process.

- Extensive experiments demonstrate the effectiveness of our method, achieving a remarkable size reduction of **over 58×** compared to vanilla 3DGS on Mip-NeRF360 Barron et al. (2022). Even compared with the SOTA compression method, we still achieves a **16.91%** bit rate savings, while preserving or improving rendering quality.

## 2 RELATED WORK

### 2.1 3D GAUSSIAN SPLATTING

3D Gaussian Splatting (3DGS) Kerbl et al. (2023) represents 3D scenes using a large number of anisotropic Gaussian ellipsoids. With high-fidelity and real-time rendering, 3DGS has rapidly surpassed NeRF-based methods Mildenhall et al. (2021); Li et al. (2023); Müller et al. (2022) and becomes a popular paradigm in the field of Novel View Synthesis (NVS). Subsequent works improve 3DGS from various perspectives. Lu et al. (2024b); Zhang et al. (2024); Liang et al. (2024); Ye et al. (2024); Kheradmand et al. (2025); Rota Bulò et al. (2024) are devoted to improving the rendering quality, while others focus on accelerating training Fang & Wang (2024b); Mallick et al. (2024) and reducing storage costs Chen et al. (2024b); Fan et al. (2024). For Level-of-Detail (LoD) management, hierarchical and spatial partitioning strategies have been introduced to enable scalable rendering Ren et al. (2024); Kerbl et al. (2024). Moreover, this promising representation has been explored in broader domains such as dynamic scene reconstruction Wu et al. (2025; 2024), 3D content generation Tang et al. (2023), human pose estimation Qian et al. (2024); Zheng et al. (2024), and generalizable Gaussian Splatting Liu et al. (2024b); Charatan et al. (2024). However, 3DGS requires storing a large number of primitives and attributes, leading to significant storage and bandwidth overhead. Efficient 3DGS compression is crucial for the further deployment in real-world applications.

### 2.2 3DGS COMPRESSION

Existing 3DGS compression methods can be broadly classified into the following categories: **(a) Compact structure.** Scaffold-GS Lu et al. (2024b) introduces anchors to aggregate local Gaussians, leading to a compact structure and inspiring many follow-up works Chen et al. (2024b; 2025); Wang et al. (2024). Alternative compact representations include structuring Gaussian attributes as compressible images Zhang et al. (2025) or planes Lee et al. (2025), and employing hierarchical structures Ren et al. (2024); Wang et al. (2024); Kerbl et al. (2024). **(b) Gaussian pruning.** Lee et al. (2024) learn a binary mask to prune Gaussians with negligible impact on rendering. Mallick et al. (2024); Fang & Wang (2024a) proposes an importance metric for Gaussian densification and pruning. Alternative approaches focus on pruning color attributes Zhang et al. (2023); Fan et al. (2024). **(c) Quantization.** Vector Quantization (VQ) and Residual Vector Quantization (RVQ) with codebooks are used for compressing Gaussian attributes Lee et al. (2024); Navaneet et al. (2024); Zhang et al. (2023); Papantonakis et al. (2024); Fan et al. (2024); Girish et al. (2024). Besides, Chen et al. (2024b; 2025) employ an adaptive quantization strategy, in which each attribute is assigned a learnable quantization step. **(d) Entropy coding.** HAC Chen et al. (2024b) designs an entropy model to model the probability distribution of anchor attributes, enabling efficient entropy coding. Some works enhance entropy model by incorporating additional prior, including hyper-priors Liu et al. (2024a), spatial Wang et al. (2024) and channel-wise auto-regression Chen et al. (2025). Despite the

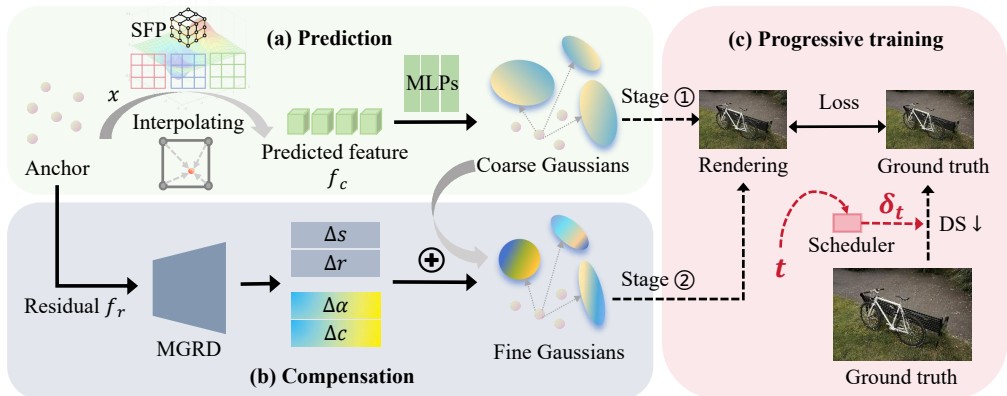

Figure 2: Pipeline of **Predict-GS**. (**a**) Each anchor queries the Spatial Feature Pool (SFP) based on its location to retrieve a predicted feature $f_c$, which is then used to infer coarse Gaussians following the formulation in Scaffold-GS Lu et al. (2024b). (**b**) To refine the Gaussians, the proposed residual for each anchor is fed into a Multi-head Gaussian Residual Decoder (MGRD), which produces fine-grained corrections to both shape and appearance of predicted Gaussians, obtaining refined, high-fidelity ones. (**c**) The reconstruction process is divided into two stages: the first stage focuses solely on optimizing the Spatial Feature Pool (SFP) for prediction, while the second stage incorporates residual compensation to progressively refine the predicted Gaussians. A cosine-scheduled downsampling (denoted as DS ↓) is performed to training views, enabling a smooth coarse-to-fine fitting.

progress, a notable limitation persists—existing methods independently encode each primitive in its entirety, overlooking the spatial redundancy inherent among them. Addressing this issue is the core motivation behind our predictive compression framework.

## 3 METHODOLOGY

### 3.1 OVERVIEW

Our overall framework is illustrated in Fig. 2. We first introduce our predictive compression framework in Sec. 3.2, which includes the Spatial Feature Pool (SFP) for prediction and a residual compensation module based on a Multi-head Gaussian Residual Decoder (MGRD). Then, in Sec. 3.3 we present a progressive training strategy tailored-designed for the predictive framework. Finally, Sec. 3.4 details the training loss and the coding pipeline.

### 3.2 PREDICTIVE FRAMEWORK

**Preliminary and notation.** Anchor-based Gaussian Lu et al. (2024b) adopts anchors as spatial primitives. Each anchor encodes position $x$, feature $f$, scale $l$, and offsets $o$. A set of associated 3D Gaussians which possesses attributes including position $\mu$, opacity $\alpha$, color $c$, scale $s$ and rotation $r$ is derived from each anchor. Specifically, each Gaussian position $\mu$ is derived from the anchor's base position $x$, scale $l$, and offsets $o$, while the remaining attributes $\alpha$, $c$, $s$, $r$ are inferred via MLPs that take anchor feature $f$, relative viewing distance $\delta_c$ and viewing direction $\vec{d_c}$ as input. More details are in Appendix A.11.

**Prediction.** In anchor-based Gaussian Lu et al. (2024b), each anchor is associated with a high-dimensional feature $f$, which accounts for nearly 60% of the overall model size, as shown in Tab. 3. However, owing to the spatial continuity of the scene, allocating a distinct feature to each anchor leads to significant spatial redundancy. To address the limitation, we propose a Spatial Feature Pool (SFP) for compact storage of shared spatial features, allowing multiple anchors to access a shared feature for Gaussian prediction. Inspired by Instance-NGP Müller et al. (2022) and Compressed NeRF Chen et al. (2024a), our SFP is built on binarized multi-resolution 3D hash grids and 2D hash planes for the following considerations. (**a**) Binarization ensures that the SFP remains lightweight. (**b**) Multi-resolution design allows it to adaptively capture and fuse spatial features at different scales. To

balance spatial fidelity and memory efficiency, we leverage low-resolution 3D grids to encode coarse volumetric structures, thereby avoiding the prohibitive memory cost of dense high-resolution 3D volumes. Meanwhile, high-resolution 2D planes are utilized to capture fine-grained, high-frequency details with minimal overhead. **(c)** The use of hash tables not only enables efficient feature querying, but also reduces redundancy among anchors, since anchor coordinates with similar spatial feature may be mapped to the same entry. Specifically, rather than storing a feature $f$ for each anchor in an explicit manner, we query the corresponding spatial feature $f_c$ from SFP using anchor's position $x$.

$$f_{\mathrm{xyz}} = \mathcal{I}(x, H_{\mathrm{xyz}}), \quad \{f_{\mathrm{xy}}, f_{\mathrm{xz}}, f_{\mathrm{yz}}\} = \{ \mathcal{I}(x, H_{\mathrm{xy}}), \mathcal{I}(x, H_{\mathrm{xz}}), \mathcal{I}(x, H_{\mathrm{yz}}) \}, \tag{1}$$

$$f_c = F(concat(f_{\mathrm{xyz}}, f_{\mathrm{xy}}, f_{\mathrm{xz}}, f_{\mathrm{yz}})), \tag{2}$$

where $\mathcal{I}$ represents the interpolation operation Chen et al. (2024a), $H_{\mathrm{xyz}}$ represents the 3D Hash grids and $H_{\mathrm{xy}}, H_{\mathrm{xz}}, H_{\mathrm{yz}}$ denotes the three 2D planes of the decomposition. $F(\cdot)$ is a MLP for multi-scale feature fusion. In replace of the original feature $f$, the compact shared feature $f_c$ from SFP is used to infer the attributes of $k$ Gaussians via MLP.

$$\{s_i, r_i, \alpha_i, c_i\}_{i=0}^{k-1} = \mathrm{MLP}(f_c, \delta_c, \vec{d}_c). \tag{3}$$

Note that Chen et al. (2024b) employs a similar grid structure as our SFP in its entropy model. In contrast, its grid is used to model the probability of attributes for entropy coding in post-processing, whereas our SFP differs fundamentally by directly predicting features during reconstruction stage.

**Residual compensation.** However, relying solely on the SFP to predict 3D Gaussians often leads to degraded rendering quality, as the interpolated features $f_c$ are obtained through continuous sampling over spatial coordinates. This interpolation inherently smooths out anchor-specific signals, compromising the ability to capture localized high-frequency details around each anchor. To address this limitation, we introduce a residual feature $f_r$ for each anchor to explicitly compensate for the lost spatial specificity. The residual $f_r$ is decoded by a Multi-head Gaussian Residual Decoder (MGRD) to produce attribute corrections for Gaussian refinement. In details, the shape head $\mathrm{MGRD}_s$ models variations for scale and rotation ($\Delta s, \Delta r$) of Gaussians, while the appearance head $\mathrm{MGRD}_a$ captures residuals for opacity and color ($\Delta \alpha, \Delta c$). $\delta_c, \vec{d}_c$ provide view-dependent priors.

$$\{\Delta s_i, \Delta r_i\}_{i=0}^{k-1} = \mathrm{MGRD}_s(f_r, \delta_c, \vec{d}_c), \quad \{\Delta \alpha_i, \Delta c_i\}_{i=0}^{k-1} = \mathrm{MGRD}_a(f_r, \delta_c, \vec{d}_c), \tag{4}$$

The predicted Gaussians are refined by incorporating the residual corrections into their respective attributes, resulting in more precise representations. This residual-guided enhancement transforms coarse predicted Gaussians into fine-grained ones, which are subsequently used for splatting and synthesize high-fidelity novel views.

$$\{\hat{s}_i, \hat{r}_i, \hat{\alpha}_i, \hat{c}_i\}_{i=0}^{k-1} = \{s_i, r_i, \alpha_i, c_i\}_{i=0}^{k-1} + \{\Delta s_i, \Delta r_i, \Delta \alpha_i, \Delta c_i\}_{i=0}^{k-1}. \tag{5}$$

where $\{\hat{s}, \hat{r}, \hat{\alpha}, \hat{c}\}$ is the refined attributes of Gaussians.

### 3.3 Prediction-tailored progressive training

**Two-stage training.** To better align with our proposed predictive compression framework, we adopt a two-stage training strategy. In the first stage, the residual compensation module is disabled, encouraging the SFP to concentrate on learning precise Gaussian prediction. In the second stage, we activate both the residual attribute and MGRD, enabling the model to learn corrective signals that refine the predictions, and recover high-frequency details in the scene.

**Progressive downsampling schedule.** As demonstrated in the ablation study (Tab. 4.3), our predictive compression framework is not well supported by the conventional 3DGS training pipeline, which typically optimizes models using high-resolution views throughout the training process. High-resolution views introduce excessive high-frequency details that exceed the representational capacity of our binarized SFP during the initial training stage. As a result, the SFP tends to overfit view-specific details while failing to capture the global structure and underlying geometry.

To address this issue, we propose a progressive training strategy tailored to our predictive compression framework. Specifically, we train the Spatial Feature Pool (SFP) using downsampled views obtained via Bilinear Interpolation $BI(\cdot)$, which offers smoother transitions and better structural preservation

compared to pooling-based methods. Downsampling reduces high-frequency signals in the training views and mitigates overfitting to view-specific high-frequency details, encouraging the SFP to focus on the global structure of the scene. To progressively restore the resolution of training views, the downsampling scale is gradually increased during training according to a cosine annealing schedule, which facilitates stable and effective optimization. Ablation studies about the schedule are presented in Appendix A.6. For each training iteration $t$, the downsampling scale $\delta_t$ is derived as follows:

$$\delta_t = \delta_e + 0.5 * (\delta_s - \delta_e) * (1 + cos(\pi * t/T)) \quad (6)$$

where $\delta_s, \delta_e$ represent the initial and final scale, $T$ denotes total iterations using downsampling. For iteration $t$, the ground truth $\hat{I}_{gt}$ used for training is downsampled from original image $I_{gt}$, according to the scale $\delta_t$:

$$\hat{I}_{gt} = BI(I_{gt}, \delta_t). \quad (7)$$

The scale $\delta_t$ approaches $\delta_e = 1.0$ at predefined iteration $T$, after which the model is consistently trained with full-resolution images for fine-tuning. This progressive strategy not only addresses the suboptimal rendering quality but also enhances stability during the transition to high-resolution views for residual learning.

### 3.4 TRAINING AND CODING

**Entropy model and training loss.** We adopt the Entropy Model $EM$ in HAC Chen et al. (2024b) but modify its output layer adapting to our proposed residual $f_r$. For each anchor $x$, the probability distributions and quantization steps of its attributes $a \in \{f_r, l, o\}$ are estimated by $EM$. Each anchor attribute $a$, along with its associated probability $p(a)$ and quantization step $q_a$, is passed to the Arithmetic Encoder (AE) for entropy coding. The detailed derivation of the probability can be found in Appendix A.3. The training objective $L$ remains the Rate-Distortion (R-D) loss as in HAC Chen et al. (2024b), which jointly constrains the rendering quality and the entropy of the attributes. The loss is formulated as:

$$L = L_{scaffold} + \lambda_e L_{entropy} + \lambda_m L_m. \quad (8)$$

$L_{scaffold}$ denotes the loss function definded in Scaffold-GS Lu et al. (2024b) for higher rendering quality, while $L_{entropy}$ and $L_m$ represent the entropy loss and mask loss of HAC Chen et al. (2024b) for lower bit rate. **Note that $L_{entropy}$ includes the entropy loss of our proposed SFP and $f_r$.** $\lambda_e$ and $\lambda_m$ are pre-defined hyper-parameters. More details about the loss can be found in Appendix A.4.

**Encoding and decoding.** In encoding phase, anchor positions $x$ are used to query Entropy Model $EM$ Chen et al. (2024b), yielding probability distributions and adaptive quantization steps for anchor attributes. These, along with anchor attributes $l, o, f_r$, are fed into the Arithmetic Encoder (AE) for entropy coding. $x$ are quantized to 16 bits and stored directly. Additionally, Spatial Feature Pool (SFP), Multi-head Gaussian Residual Decoder (MGRD), $EM$, the mask Chen et al. (2024b), and other MLPs are serialized into bit stream for storage and transmission. In decoding phase, $x$, SFP, MGRD, $EM$, mask, and MLPs are first restored. $x$ then query $EM$ to retrieve the corresponding distributions and quantization steps for attributes, which are used by the Arithmetic Decoder (AD) to reconstruct residuals $f_r$, scales $l$, and offsets $o$. Complete encoding and decoding procedures are detailed in Appendix A.7.

Table 1: Quantitative results on Mip-NeRF360 and Tank&Temples.

| Dataset | | Mip-NeRF360 | | | Tank&Temples | | | |
|---|---|---|---|---|---|---|---|---|
| Method \ Metrics | PSNR↑ | SSIM↑ | LPIPS↓ | SIZE↓ | PSNR↑ | SSIM↑ | LPIPS↓ | SIZE↓ |
| 3DGS | 27.21 | 0.815 | 0.214 | 734MB | 23.14 | 0.841 | 0.183 | 411MB |
| Scaffold-GS | 27.50 | 0.806 | 0.252 | 253.9MB | 23.96 | 0.853 | 0.177 | 86.5MB |
| Compact3DGS | 27.03 | 0.797 | 0.247 | 29.1MB | 23.32 | 0.831 | 0.202 | 20.9MB |
| Compressed3D | 26.98 | 0.801 | 0.238 | 28.80MB | 23.32 | 0.832 | 0.194 | 17.28MB |
| LightGaussian | 27.13 | 0.806 | 0.237 | 45MB | 23.44 | 0.832 | 0.202 | 25MB |
| EAGLES | 27.23 | 0.810 | 0.240 | 54MB | 23.37 | 0.840 | 0.200 | 29MB |
| CompGS | 27.26 | 0.800 | 0.240 | 16.50MB | 23.70 | 0.840 | 0.210 | 9.60MB |
| Compact3D | 27.12 | 0.806 | 0.240 | 19MB | 23.44 | 0.838 | 0.198 | 13MB |
| **HAC** | 27.53 | 0.807 | 0.238 | 15.26MB | 24.04 | 0.846 | 0.187 | 8.10MB |
| **Ours** | 27.53 | 0.804 | 0.246 | 12.68MB | 24.18 | 0.848 | 0.194 | 6.62MB |
| **Ours (high rate)** | 27.78 | 0.809 | 0.236 | 19.28MB | 24.20 | 0.853 | 0.185 | 9.32MB |

Table 2: Quantitative results on Deep Blending.

| Dataset Method \ Metrics | Deep Blending PSNR↑ | SSIM↑ | LPIPS↓ | SIZE↓ |
|---|---|---|---|---|
| 3DGS | 29.41 | 0.903 | 0.243 | 676MB |
| Scaffold-GS | 30.21 | 0.906 | 0.254 | 66MB |
| Compact3DGS | 29.73 | 0.900 | 0.258 | 23.8MB |
| Compressed3D | 29.38 | 0.898 | 0.253 | 25.30MB |
| LightGaussian | — | — | — | — |
| EAGLES | 29.86 | 0.910 | 0.250 | 52MB |
| CompGS | 29.69 | 0.900 | 0.280 | 8.77MB |
| Compact3D | — | — | — | — |
| **HAC** | 29.98 | 0.902 | 0.269 | 4.35MB |
| **Ours** | 30.17 | 0.906 | 0.271 | 3.75MB |
| **Ours (high rate)** | 30.49 | 0.911 | 0.260 | 6.35MB |

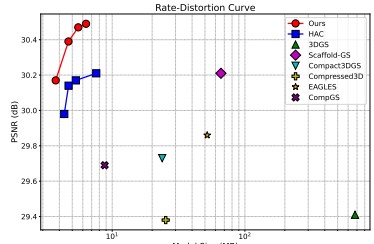

Figure 3: Rate-Distortion (R-D) curve on Deep Blending dataset. A point closer to the top-left corner indicates better compression performance.

Table 3: Bit allocation comparison between HAC and ours on datasets Mip-NeRF360, Tank&Temples and Deep Blending. **Feature** (in MB) denotes the bit rate of anchor features (i.e., $f$ in HAC, and SFP together with $f_r$ in ours). **BSF** indicates the **B**it rate **S**avings of anchor **F**eatures. **Others** (in MB) refers to the bit rate of all remaining components.

| Dataset Method \ Metrics | Mip-NeRF360 Feature | BSF | Others | Tank&Temples Feature | BSF | Others | Deep Blending Feature | BSF | Others |
|---|---|---|---|---|---|---|---|---|---|
| **HAC** | 5.92 | -0% | 9.45 | 2.95 | -0% | 5.17 | 1.47 | -0% | 3.11 |
| **Ours** | 3.66 | **-38.18%** | 9.02 | 1.93 | **-34.58%** | 4.70 | 0.78 | **-46.94%** | 2.97 |

# 4 EXPERIMENT

## 4.1 EXPERIMENTAL SETTINGS

**Implementation details.** We build our **Predict-GS** based on the official implementation of HAC Chen et al. (2024b). The SFP is implemented by 3D hash grids with 12 resolution levels and 2D hash planes with 4 resolution levels. Dimension of predicted feature $f_c$ is set as 50 to be consistent with original feature dimension in HAC Chen et al. (2024b). The residual dimension is set to 25, and the corresponding ablation study can be found in Appendix 6. Our MGRD is implemented by a two-layer MLP with ReLU activation. We pretrain the SFP for 5k iterations. The residual branch is activated at iteration 5k and the entropy loss is incorporated at iteration 10k. The total number of iterations is 30k, which is the same as in HAC Chen et al. (2024b). More details can be found in Appendix A.2 and A.4. All experiments are performed on a NVIDIA A100 GPU.

**Datasets, metrics and baselines.** Evaluation is performed on three public benchmarks: Mip-NeRF360 Barron et al. (2022), Tank&Temples Knapitsch et al. (2017) and Deep Blending Hedman et al. (2018). Quantitative results are reported using PSNR (dB), SSIM, and LPIPS for reconstruction quality, and model size (MB) for compression efficiency. We compare our predictive framework

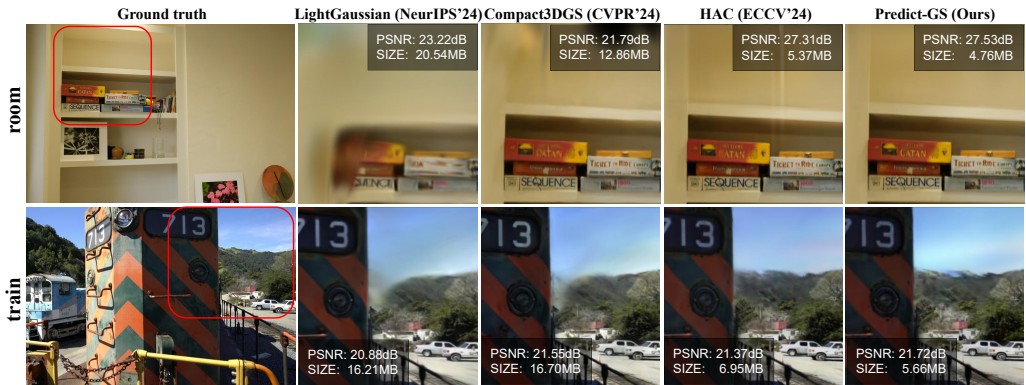

Figure 4: Qualitative comparison. In each row, the leftmost image shows the ground truth. The red boxed area is magnified to facilitate more detailed comparison. PSNR and model size are provided.

with 3DGS Kerbl et al. (2023), Scaffold-GS Lu et al. (2024b) and representative 3DGS compression methods (Compact3DGS Lee et al. (2024), Compressed3D Niedermayr et al. (2024), LightGaussian Fan et al. (2024), EAGLES Girish et al. (2024), CompGS Liu et al. (2024c), Compact3D Navaneet et al. (2024), HAC Chen et al. (2024b)), which encompass various compression techniques as discussed in Sec 2.2. It is worth mentioning that HAC Chen et al. (2024b) serves as a strong baseline and is widely acknowledged as the state-of-the-art (SOTA) in 3DGS compression.

## 4.2 EXPERIMENTAL RESULTS

**Quantitative and qualitative comparison.** A quantitative comparison between our method and representative 3DGS compression approaches is presented in Tab. 1 and Tab. 2, where the top-1, top-2, and top-3 for each metric are highlighted in red, orange, and yellow respectively. In terms of compression performance, our method **(Ours)** achieves significant bit rate savings over existing 3DGS compression methods. Notably, even compared to the SOTA HAC Chen et al. (2024b), our method still yields an average bit rate reduction of 16.91% on Mip-NeRF360 Barron et al. (2022) and 18.27% on Tank&Temples Knapitsch et al. (2017), respectively, which is a significant gain for compression tasks. This gain directly stems from the removal of spatial redundancy among anchor features, validating the effectiveness of our predictive compression framework. For rendering quality, **Ours** outperforms most 3DGS compression baselines and achieves competitive rendering quality with SOTA method HAC Chen et al. (2024b), even with 0.1–0.2 dB improvements in PSNR on Tank&Temples Knapitsch et al. (2017) and Deep Blending Hedman et al. (2018). We attribute these gains to the synergy between our improved progressive training strategy and prediction-driven compression, as presented in Tab. 4. Additionally, a high rate model **Ours (high rate)** is also included to showcase the maximum for reconstruction quality, in a storage-unconstrained setting. We presents a visual comparison of rendering quality between our method and representative compression methods. As shown in Fig. 4, our method achieves comparable rendering quality with HAC Chen et al. (2024b) and outperforms LightGaussian Fan et al. (2024) and Compact3DGS Lee et al. (2024), while maintaining the lowest bit rate among all. More visual examples are provided in Fig. 10 of Appendix A.8.

**R-D performance.** R-D curve provides a clear visualization of the trade-off between bit rate and reconstruction quality—points closer to the top-left corner indicate better compression performance. We compare our method against several representative compression methods on Deep Blending Hedman et al. (2018) dataset. As shown in Fig. 3, our method achieves superior rate-distortion performance, indicating that under the same bit rate constraints, our model delivers higher reconstruction quality. Conversely, for the same level of rendering quality, our method requires a lower bit rate.

**Bit allocation.** To further confirm the source of our compression gains and highlight the effectiveness of our method, we report bit rate allocation on each dataset compared with SOTA HAC Chen et al. (2024b). As shown in Tab. 3, the majority of bit rate reduction stems from eliminating spatial redundancy among anchor feature. **The direct gain accounts for up to 40% on average relative to the bit rate of anchor feature.** (We reproduce HAC Chen et al. (2024b) using its official code to obtain the results in Tab. 3.)

## 4.3 ABLATION STUDIES

**Effectiveness of each component.** Tab. 4 presents quantitative results of our component-wise ablation study. **Baseline** is the SOTA HAC Chen et al. (2024b). **Ours w/ SFP** shows the results using only SFP prediction, **Ours w/ SFP w/ $f_r$** includes the additional residual branch, and **Ours all** further incorporates our progressive training strategy, representing the full method. Compared to **Baseline**, **Ours w/ SFP** achieves a significant reduction in model size because only a lightweight SFP needs to be stored, instead of the complete feature for each anchor. Notably, **Ours w/ SFP** achieves a PSNR of 22.61dB on Tank&Temples Knapitsch et al. (2017) dataset, indicating that SFP possesses the capability to capture spatial features for 3D scene reconstruction and shows strong potential for Gaussian prediction. **Ours w/ SFP w/ $f_r$** highlights the effectiveness of residual compensation for fidelity in local details. While albeit with additional bit rate overhead, **Ours w/ SFP w/ $f_r$** still achieves a lower overall bit rate compared to **Baseline**. **Ours all** further enhances the rendering quality compared to **Ours w/ SFP w/ $f_r$** (0.17dB ↑ on Tank&Temples Knapitsch et al.

Table 4: Ablation study on dataset Tank&Temples Knapitsch et al. (2017). **w/ SFP** denotes prediction with SFP. **w/ $f_r$** and **w/ PT** represent the residual compensation and progressive training respectively.

| Method | w/ SFP | w/ $f_r$ | w/ PT | PSNR↑ | SSIM↑ | LPIPS↓ | SIZE↓ |
|---|---|---|---|---|---|---|---|
| Baseline | — | — | — | 24.04 | 0.846 | 0.187 | 8.10MB |
| Ours w/ SFP | ✓ | | | 22.61 | 0.802 | 0.261 | 4.75MB |
| Ours w/ SFP w/ $f_r$ | ✓ | ✓ | | 24.01 | 0.841 | 0.200 | 6.75MB |
| Ours all | ✓ | ✓ | ✓ | 24.18 | 0.848 | 0.194 | 6.62MB |

(2017)), demonstrating the effectiveness of our progressive training in maximizing the potential of our predictive framework.

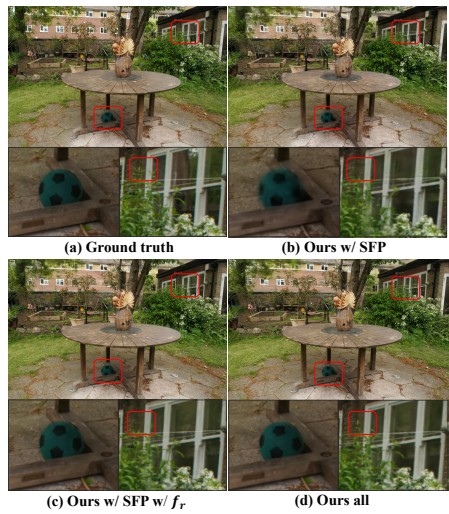

(a) Ground truth      (b) Ours w/ SFP

(c) Ours w/ SFP w/ $f_r$      (d) Ours all

Figure 5: Visualization for ablation study.

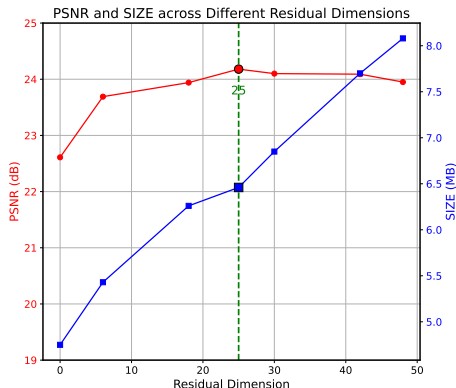

Figure 6: Ablation study on residual dimension. The experiment is conducted on the Tank&Temples Knapitsch et al. (2017) dataset. We report results of average PSNR and model size across the entire dataset under different residual dimensions.

**Visualization.** We provide visualization in Fig. 5 on the *garden* scene from Mip-NeRF360 Barron et al. (2022). Regions with noticeable differences are highlighted. As shown in Fig. 5, **Ours w/ SFP** indicates that predicted Gaussians can roughly fit the 3D scene context but fail to capture fine details. **Ours w/ SFP w/ $f_r$** demonstrate that residual compensation can effectively correct the inaccurate predictions (e.g., blurriness and distortions near the football and plants). **Ours all** further shows that the progressive training strategy leads to additional improvements in rendering quality.

**The dimension of residual.** We set residual dimension as 25 for default. Fig. 6 provides ablation study on Tank&Temples Knapitsch et al. (2017) to validate the effectiveness of our choice. When the dimension exceeds 25, the bit rate increases significantly while rendering quality remains unchanged or even decreased (too much channel redundancy may hinder optimization), indicating that 25-dimension is sufficient to compensate for the bias of the predicted Gaussians. In contrast, using dimension lower than 25 may reduce bit rate, but leads to a notable drop in quality. These results suggest that a 25-dimensional residual strikes a good balance, providing sufficient capacity to recover missing details without introducing excessive channel redundancy.

## 5 CONCLUSION

In this work, we introduce prediction and residual compensation into 3DGS compression, and propose a prediction-driven compression framework equipped with a tailored progressive training strategy, effectively mitigating spatial redundancy across anchors. Extensive experiments on public benchmarks confirm the effectiveness of our framework and individual contribution of each component.

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

# A APPENDIX

## A.1 OVERVIEW

This appendix provides supplementary materials and more details to support the main paper. The contents are organized as follows:

- A.2 More implementation details.
- A.3 Entropy model.
- A.4 Training loss and training details.
- A.5 Efficiency analysis.
- A.6 Ablation studies on training scheduling.
- A.7 Encoding and decoding pipeline and visualization.
- A.8 Visually comparison with representative 3DGS compression works.
- A.9 Results for each scene.
- A.10 Limitation and discussion.
- A.11 Brief overview of 3DGS and anchor-based variant.

## A.2 MORE IMPLEMENTATION DETAILS

We build our predictive compression framework based on the official implementation of HAC Chen et al. (2024b). In our implementation, we adhere to the same data processing protocols, model optimization strategies, and hyperparameter settings as those employed in HAC Chen et al. (2024b) for a fair comparison.

**Prediction and residual compensation.** We construct our Spatial Feature Pool (SFP) using binarized multi-resolution 3D hash grids and 2D planes. The resolution list of 3D grids is (18, 24, 33, 44, 59, 80, 108, 148, 201, 275, 376, 514), while resolution list of 2D planes is (130, 258, 514, 1026). Each vertex in the grids or the planes is individually mapped to an entry in hash table. The hash table for the grids has a size of $2^{13}$, while the hash table for the planes have a size of $2^{15}$. Each entry in the hash tables have a 4-dimension feature. The fusion network following the SFP is a two-layer MLP, with the output dimension set to 50, which is consistent with the original anchor feature dimension used in HAC Chen et al. (2024b). The residual dimension for each anchor is set to 25, which is empirically determined to be optimal through ablation studies. Larger dimensions lead to a rapid increase for bit rate, while smaller ones result in significant quality degradation. The Multi-head Gaussian Residual Decoder (MGRD) is implemented as a two-layer MLP with ReLU activations. The output dimension of the shape head is 7, matches that of the Gaussian attributes scale (3) and rotation (4). The output dimension of the appearance head is 4, matches that of the Gaussian attributes opacity (1) and color (3).

**Prediction-tailored progressive training.** We adopt a customized progressive training strategy, starting from iteration 0 and continuing for $T =$9k iterations. The initial downsampling scale $\delta_s$ is set to 0.7 and gradually increases to $\delta_e$ 1.0 following a cosine schedule. During the first 5k iterations of training, we disable the residual branch and exclusively train the SFP. After 5k iterations, we enable the residual compensation.

## A.3 ENTROPY MODEL

We adopt the Entropy Model $EM$ as in HAC Chen et al. (2024b) but modify its output layer adapting to our proposed residual $f_r$. For each anchor $x$, the probability distribution of its attributes $a$ is modeled as a Gaussian distribution $a \sim N(\mu_a, \theta_a)$, with both the mean $\mu_a$ and variance $\theta_a$ estimated by $EM$.

$$\{\mu_a, \theta_a, q_a\} = EM(x), \quad a \in \{f_r, o, l\}, \tag{9}$$

where $q_a$ is the corresponding quantization step for $a$. For each anchor attribute $a$, its probability $p(a)$ is computed using the probability density function of a Gaussian distribution $\phi_{\mu_a, \theta_a}(\cdot)$.

$$p(a) = \int_{a-\frac{1}{2}q_a}^{a+\frac{1}{2}q_a} \phi_{\mu_a, \theta_a}(x)dx = \Phi_{\mu_a, \theta_a}(a + \frac{1}{2}q_a) - \Phi_{\mu_a, \theta_a}(a - \frac{1}{2}q_a). \tag{10}$$

where $\Phi(\cdot)$ denotes the Cumulative Distribution Function (CDF) of $\phi(\cdot)$. Each anchor attribute $a$, along with its associated probability $p(a)$ and quantization step $q_a$, is passed to the Arithmetic Encoder (AE) for entropy coding.

### A.4 TRAINING LOSS AND TRAINING DETAILS

The training objective $L$ in HAC Chen et al. (2024b) consists of three parts, which jointly constrains the rendering quality and the entropy of the attributes. The loss is formulated as:

$$L = L_{scaffold} + \lambda_e L_{entropy} + \lambda_m L_m. \tag{11}$$

$L_{scaffold}$ denotes reconstruction loss definded in Scaffold-GS Lu et al. (2024b), including L1 loss and SSIM loss between ground truth $I_{gt}$ and rendered image $I_{render}$, as well as a regularization term to encourage smaller scale of Gaussians.

$$L_{scaffold} = L_1(I_{gt}, I_{render}) + \lambda_{SSIM}L_{SSIM}(I_{gt}, I_{render}) + \lambda_{reg}L_{reg}, \tag{12}$$

$L_{entropy}$ and $L_m$ represent the entropy loss and mask loss definded in HAC Chen et al. (2024b). $\lambda_e$ is set to 0.004 and $\lambda_m$ is set to 5e-4 for default in our evaluation. Notably, $\lambda_e$ is set to 0.0005 to attain a high rate model **Ours (high rate)**.

$$L_{entropy} = \sum_{a \in \{f_r, l, o\}} \sum_{i=1}^{N} \sum_{j=1}^{D} (-\log_2 p(a_{i,j})) + L_{hash}, \tag{13}$$

$$L_{hash} = M_+ \times (-\log_2(h_f)) + M_- \times (-\log_2(1 - h_f)). \tag{14}$$

where N is the number of anchors and $a_{i,j}$ is $a_i$'s j-th dimension value. Minimizing the entropy loss encourages a high probability estimation for $p(a_{i,j})$, guiding the accurate estimation for $\mu_{i,j}, \theta_{i,j}$ of the Entropy Model $EM$. $L_{hash}$ quantifies the entropy of the binarized embeddings in the hash tables employed by our proposed SFP and $EM$ of Chen et al. (2024b), where $M_+$ and $M_-$ are total numbers of $+1$ and $-1$, and $h_f$ is the occurrence frequency of the symbol $+1$. The concept of mask loss $L_m$ was originally proposed in Compact3DGS Lee et al. (2024) to promote the pruning of redundant Gaussians and anchors during training.

The total number of training iterations is 30k. During the first 5k iterations, we only train the SFP module to fit the basic structure of the scene. After 5k iterations, the residual compensation branch is introduced. Additionally, during the first 10k iterations, the entropy loss constraint $L_{entropy}$ is disabled to allow the model to focus on scene reconstruction.

### A.5 EFFICIENCY ANALYSIS

**Training time and FPS.** Tab. 5 further quantify the impact on efficiency of our method. Our rendering speed experiences a slight degradation compared with HAC, due to an additional residual branch during inference. Nevertheless, our system still supports real-time rendering for 70+ FPS. Despite the residual branch in forward pass during training, the overall training time remains efficient (even 4% faster than HAC), which stems from two factors: **(a)** The entropy model becomes more lightweight, as it only estimates the probability of residual rather than the complete feature. **(b)** Our progressive training strategy avoids using high-resolution view in early training. By the way, the FPS degradation could be addressed if refined Gaussians are cached at rendering.

**Encoding and decoding time.** We also report the impact of our method on encoding and decoding time on Mip-NeRF360 dataset. As shown in Tab. 5, our method achieves an average encoding time of 6.73s and decoding time of 18.56s on the Mip-NeRF360 dataset, slightly faster than HAC. It is because our method only needs to encode a short residual, rather than lengthy feature.

Table 5: Efficiency comparison on Mip-NeRF360 (testing on a NVIDIA A100 GPU).

| Method | Size (MB)↓ | Train time (s)↓ | Encode time (s)↓ | Decode time (s)↓ | Test FPS↑ |
|--------|-----------|-----------------|------------------|------------------|-----------|
| HAC | 15.23 | 2180 | 6.96 | 18.72 | 87.88 |
| Ours | 12.65 | 2097 | 6.73 | 18.56 | 71.74 |

Table 6: Results of different annealing schedules. **Linear** denotes that the downsampling scale is annealing linearly during the progressive training process, while **Ours (Cosine)** uses a cosine-based annealing schedule. **Non** refers to the case without progressive annealing, meaning a fixed initial downsampling scale is applied throughout SFP training and no further downsampling is applied once residual training begins.

| Dataset | | Mip-NeRF360 Barron et al. (2022) | | | |
|---------|---------|-------|-------|--------|--------|
| **Method** | **Metrics** | **PSNR↑** | **SSIM↑** | **LPIPS↓** | **SIZE↓** |
| Non | | 26.71 | 0.779 | 0.258 | 12.81MB |
| Linear | | 27.43 | 0.802 | 0.248 | 12.71MB |
| Ours (Cosine) | | 27.53 | 0.804 | 0.246 | 12.68MB |

## A.6 ABLATION STUDIES ON TRAINING SCHEDULING

**The annealing schedule.**  To fully harness the potential of our prediction-driven 3DGS compression framework, we adopt a progressive training strategy that applies cosine annealing downsampling for the training views. Tab. 6 shows the impact of various annealing schedules on the final performance of models, demonstrating the effectiveness of the cosine annealing approach we employ. As shown in the first row (**Non**), the absence of a progressive transition leads to a significant drop in quality. This is mainly due to the abrupt increase in the downsampling scale when the residual branch is introduced, which causes instability in the optimization process. The results in the second (**Linear**) and third rows (**Cosine**) demonstrate that cosine annealing outperforms linear annealing, as the smaller slope of the schedule function in the early stages facilitates a more stable training process, allowing the SFP module to effectively capture the geometry context of the scene.

**Initial downsampling scale.**  Furthermore, Tab. 7 reports the results under different initial downsampling scales, further validating the optimality of our chosen hyper-parameters. We observe that an initial downsampling scale in the range of 0.6 to 0.7 yields satisfactory results, while increasing the scale to 0.8 and over leads to a noticeable degradation in rendering quality and bit rate increasing. Considering that an excessively small initial downsampling scale would severely impair image information, while an overly large scale would diminish the benefits of progressive training, we set the initial downsampling scale to 0.7 based on ablation studies. It is worth mentioning that the hyperparameter choice generalizes well across multiple datasets.

## A.7 ENCODING AND DECODING PIPELINE AND VISUALIZATION

A visualization of the compressed bit stream is provided in the Fig. 8, where the encoding and decoding processes are annotated with step-by-step indices.

Fig.9 presents a detailed comparison of anchor feature encoding strategies between our method and other anchor-based Gaussian compression techniques.

## A.8 VISUALLY COMPARISON WITH REPRESENTATIVE 3DGS COMPRESSION WORKS

As presented in Fig 10, our method achieves superior rendering quality and lower bit rate compared to representative 3DGS compression works, LightGaussian Fan et al. (2024) and Compact3DGS Lee et al. (2024). Even when compared to the SOTA compression method HAC Chen et al. (2024b), our approach achieves a notable bit rate reduction while maintaining comparable rendering quality.

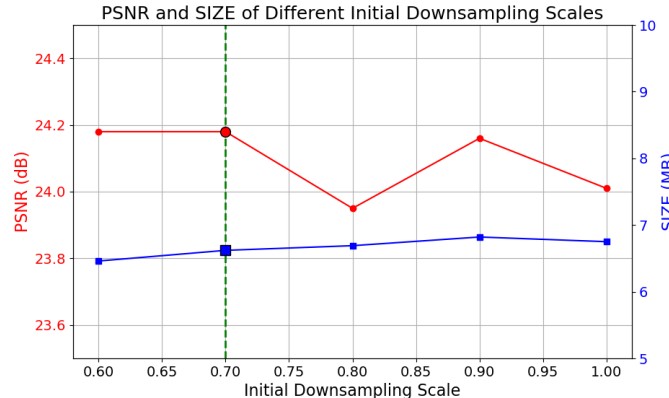

Figure 7: Ablation study on initial scale of downsampling during training. The experiment is conducted on the Tank&Temples Knapitsch et al. (2017) dataset. We report results of average PSNR and model size across the entire dataset under different initial scales of downsampling.

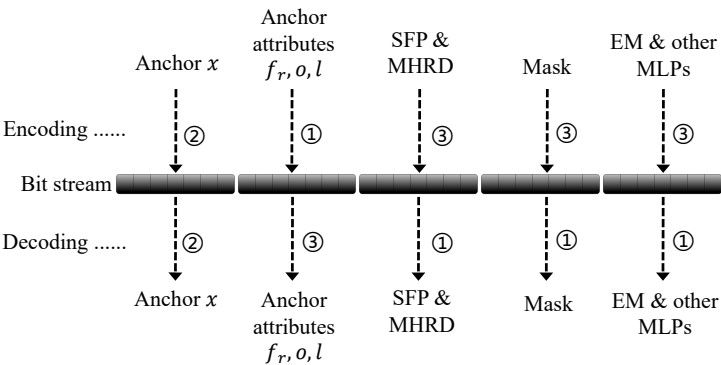

Figure 8: A visualization of the compressed bit stream and the order of encoding and decoding.

## A.9 RESULTS FOR EACH SCENE

Tab. 7, Tab. 8 and Tab. 9 present the per-scene results of our method corresponding to the comparative experiments in our paper (Tab. 1 and Tab. 2).

Table 7: Per-scene results on the Mip-NeRF360 Barron et al. (2022) dataset.

| Scene | PSNR↑ | SSIM↑ | LPIPS↓ | SIZE↓ |
|---|---|---|---|---|
| bicycle | 25.05 | 0.738 | 0.274 | 21.61 |
| garden | 27.18 | 0.837 | 0.161 | 17.23 |
| stump | 26.78 | 0.762 | 0.274 | 15.83 |
| room | 31.75 | 0.920 | 0.221 | 4.76 |
| counter | 29.30 | 0.910 | 0.201 | 6.35 |
| kitchen | 31.15 | 0.922 | 0.135 | 6.81 |
| bonsai | 32.32 | 0.943 | 0.193 | 7.90 |
| flower | 21.07 | 0.562 | 0.392 | 16.38 |
| treehill | 23.14 | 0.641 | 0.362 | 17.25 |
| **Avg** | **27.53** | **0.804** | **0.246** | **12.68** |

## A.10 LIMITATION AND DISCUSSION

**Limitation.** While our predictive compression framework reduces spatial redundancy among anchors and improves compression performance, it also increases the computational cost during training and rendering, mainly due to the additional learning of residual attributes and the residual

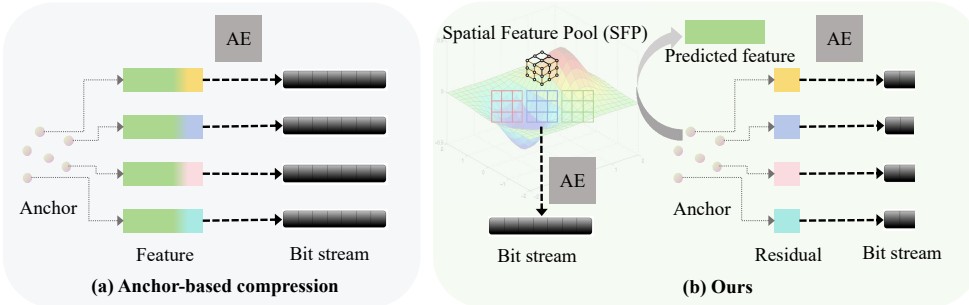

Figure 9: How to encode anchor feature. Existing anchor-based Gaussian compression encodes the complete feature for each anchor, leading to substantial bit rate overhead. Due to high similarity among neighboring anchors, a large portion of the encoded feature is redundant, as shared components are repeatedly stored (**left**). To address this, we propose a compact Spatial Feature Pool (SFP) that captures the shared feature components across anchors. Each anchor then only encodes its residual with respect to the SFP, effectively eliminating spatial redundancy and significantly reducing the bit rate (**right**).

Table 8: Per-scene results on the Tank&Temples Knapitsch et al. (2017) dataset.

| Scene | PSNR↑ | SSIM↑ | LPIPS↓ | SIZE↓ |
|-------|-------|-------|--------|-------|
| train | 22.47 | 0.816 | 0.223 | 5.66 |
| truck | 25.88 | 0.880 | 0.165 | 7.58 |
| **Avg** | **24.18** | **0.848** | **0.194** | **6.62** |

Table 9: Per-scene results on the Deep Blending Hedman et al. (2018) dataset.

| Scene | PSNR↑ | SSIM↑ | LPIPS↓ | SIZE↓ |
|-------|-------|-------|--------|-------|
| playroom | 29.65 | 0.904 | 0.269 | 4.44 |
| drjohnson | 30.69 | 0.908 | 0.273 | 3.06 |
| **Avg** | **30.17** | **0.906** | **0.271** | **3.75** |

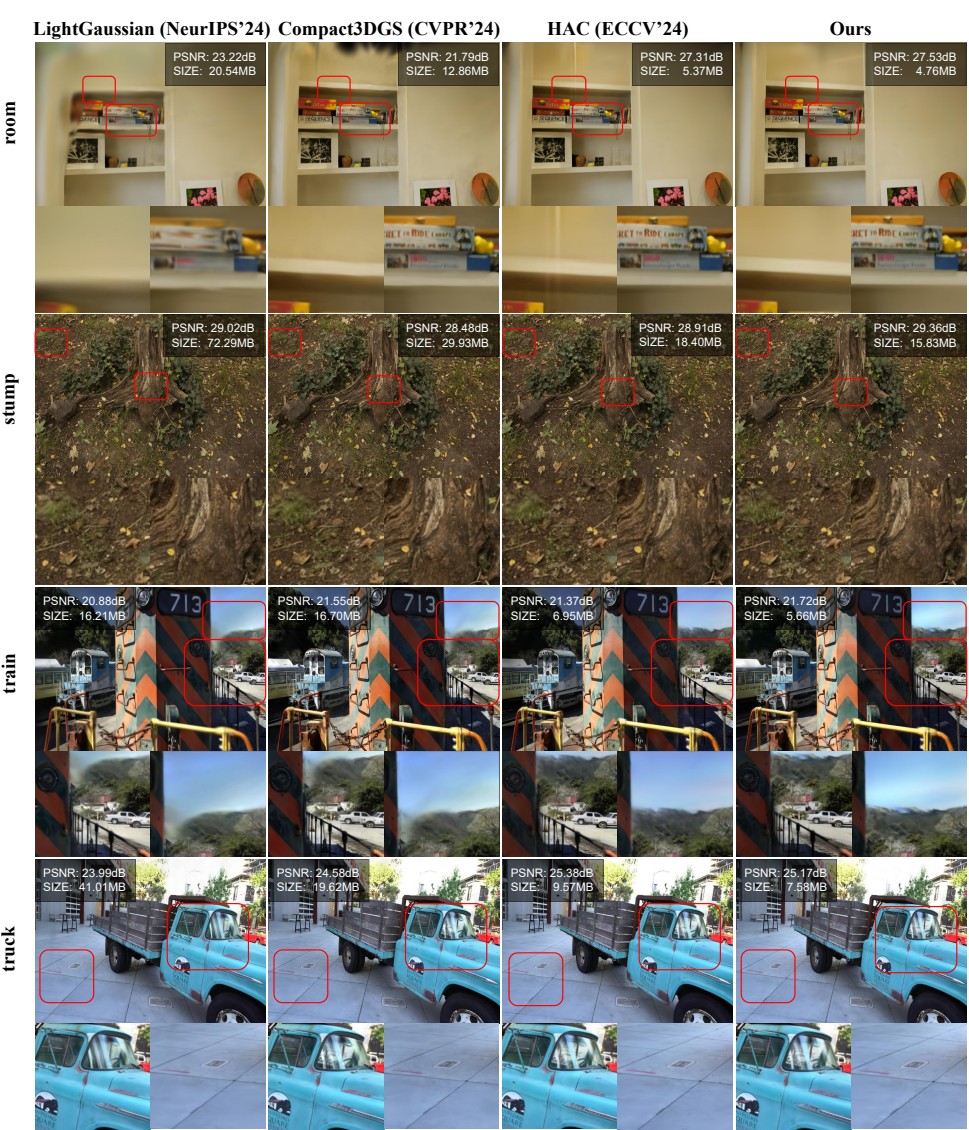

Figure 10: Visually comparison with representative 3DGS compression works. The regions with noticeable differences have been highlighted with red boxes and zoomed-in views for emphasis. We reproduce the work of LightGaussian Fan et al. (2024), Compact3DGS Lee et al. (2024) and HAC Chen et al. (2024b) to get the visual examples for qualitative visualization.

decoder (MGRD). In future work, we will explore acceleration strategies to address this issue. Further more, our method primarily targets spatial redundancy in feature attributes, while other geometric components (e.g., position, scale, offset) still contribute significantly to the bit stream. This calls for future efforts toward dedicated compression strategies for geometric attributes.

**Discussion.** We note that concurrent works HAC++ Chen et al. (2025) and HEMGS Liu et al. (2024a), improve upon HAC Chen et al. (2024b) by incorporating richer context into the entropy model, thereby achieving better compression performance. However, in this work, we still adopt HAC (Chen et al. (2024b) as our baseline and implementation framework, primarily due to its wide recognition and established effectiveness. It is worth highlighting that our contribution lies in introducing predictive techniques at the reconstruction stage to eliminate spatial redundancy — a pre-compression design. We do not change the entropy model structure of HAC (Chen et al. (2024b), which is used for post-compression process. Therefore, our method is fully compatible with the aforementioned works (Chen et al. (2025); Liu et al. (2024a), meaning that their improvements to the entropy model can be directly integrated into our predictive compression framework to further enhance overall compression performance. In addition, our work introduces the predictive coding paradigm into 3DGS compression, aligning its overall framework with that of image compression (Ma et al. (2019), and thereby enabling the adaptation of prediction-based advances from image compression to 3DGS context.

### A.11  BRIEF OVERVIEW OF 3DGS AND ANCHOR-BASED VARIANT

**3DGS.** 3DGS Kerbl et al. (2023) represents a 3D scene by an extensive number of anisotropic ellipsoids equipped with two geometry attributes (location $\mu$, covariance matrix $\Sigma$) and two appearance attributes (view-dependent Spherical Harmonic (SH) coefficients $h$ and opacity $\alpha$), which can be defined as follows:

$$\mathcal{G}(\mathbf{x}; \mu, \Sigma) = \exp\left(-\frac{1}{2}(\mathbf{x} - \mu)^T \Sigma^{-1}(\mathbf{x} - \mu)\right), \tag{15}$$

where $\mathbf{x}$ is the coordinates of a 3D scene point and the covariance matrix $\Sigma$ encodes the scale $s$ and rotation $r$ through $\Sigma = rss^T r^T$. In order to obtain images from a novel perspective, these Gaussians will be splatted to the 2D space and apply $\alpha$-blending to render the pixel value $C$:

$$C = \sum_{i \in N} c_i \alpha_i \prod_{j=1}^{i-1}(1 - \alpha_j), \tag{16}$$

where $N$ is the number of sorted Gaussians contributing to the rendering, and $c$ is the color calculated by SH coefficients $h$.

**Anchor-based variant.** Scaffold-GS Lu et al. (2024b) adopts compact anchors as spatial primitives and utilizes MLPs to infer the attributes of surrounding 3D Gaussians. Each anchor encodes properties including position $x$, feature $f$, scale $l$, and offsets $o$, from which the associated Gaussians $\mu$ are implicitly derived:

$$\{\mu_i\}_{i=0}^{k-1} = x + \{o_i\}_{i=0}^{k-1} * l, \tag{17}$$

The attributes of the $k$ neural Gaussians are inferred via dedicated MLPs that take as input the anchor feature $f$, the relative viewing distance $\delta_c$, and the viewing direction $\vec{d_c}$. For example, the opacity $\alpha$ of neural Gaussians spawned from an anchor are decoded as follows:

$$\{\alpha_i\}_{i=0}^{k-1} = \text{MLP}_\alpha(f, \delta_c, \vec{d_c}). \tag{18}$$

