# OpenReview forum: "Mitigating Spatial Redundancy: A Predictive Compression Framework for 3D Gaussian Splatting"
_ICLR.cc/2026/Conference — Submitted to ICLR 2026_

### Official Review · Reviewer_EbpM · 2025-10-26

**Soundness:** 3
**Presentation:** 3
**Contribution:** 1
**Rating:** 2
**Confidence:** 4

**Summary:**

This paper presents Predict-GS, a compression framework for 3DGS (built upon Scaffold-GS) that leverages a hash-grid–structured Spatial Feature Pool (SFP) to share and substitute features across anchors, effectively reducing spatial redundancy. To recover fine details, a Multi-head Gaussian Residual Decoder (MGRD) predicts attribute corrections from a residual feature, compensating for the over-smoothing introduced by the interpolated SFP. In addition, a progressive training schedule is introduced to stabilize optimization and improve convergence. The proposed method achieves ≈58× compression over vanilla 3DGS and ~17% bit-rate savings compared to HAC, while maintaining comparable or even superior rendering quality.

**Strengths:**

1. Compression Efficiency of Anchor Features:

Effectively enhances compression efficiency by eliminating spatial redundancy among anchor features through the predictive SFP and the MGRD refinement module.

2. Progressive Training Strategy:

Introduces a progressive training scheme that stabilizes optimization and further improves reconstruction quality.

**Weaknesses:**

1. Limited Novelty Beyond HAC:

This work is largely built upon the HAC framework, sharing similar designs for compressing offsets, scaling, residual features, and masking, with only incremental extensions through the SFP and residual refinement modules.

2. Lack of Comparison with Recent Methods:

The proposed method does not demonstrate clear advantages over recent strong baselines such as ContextGS (NeurIPS 2024), CAT-3DGS (ICLR 2025), LocoGS (ICLR 2025), HAC++, and HEMGS; in fact, some of these methods achieve better overall performance on similar benchmarks.

3. Lack of Evaluation on Large-Scale Dataset (BungeeNeRF):

The paper does not include experiments on the BungeeNeRF dataset, which is widely used in HAC, ContextGS, CAT-3DGS, and HAC++. Since this dataset represents large-scale, unbounded scenes, it is crucial to verify whether the proposed hash-grid–based SFP with limited resolution can maintain its performance under such challenging settings.

4. Increased Rendering Complexity:

Because Gaussian attribute decoding must pass through the additional MGRD for residual refinement, the rendering pipeline is more complex than Scaffold-GS, incurring extra inference computation and latency during rendering.

5. Lack of Validation under Stronger Entropy Models:

Although the paper claims compatibility with advanced entropy modeling frameworks such as HAC++ and HEMGS, it does not include experiments demonstrating whether the proposed predictive compression framework can still achieve similar gains when integrated with these stronger baselines.

**Questions:**

Please see the weakness part.

---

### Official Review · Reviewer_5VQ1 · 2025-10-29

**Soundness:** 4
**Presentation:** 3
**Contribution:** 2
**Rating:** 4
**Confidence:** 4

**Summary:**

This paper  proposes Predict-GS, a  compression framework for 3D Gaussian Splatting (3DGS). Unlike prior post-compression methods that encode every Gaussian primitive independently, Predict-GS explicitly leverages spatial redundancy among anchors through a spatial feature pool with implicit feature factorization that predicts shared scene features based on spatial coordinates. To compensate for prediction inaccuracies, the authors introduce a multi-head gaussian residual decoder that refines shape and appearance attributes. Moreover, a progressive, prediction-tailored training strategy is designed to stabilize optimization and improve rendering fidelity. Experimental results on datasets demonstrate substantial improvements.

**Strengths:**

This method achieves size reduction while preserving or even slightly improving visual quality compared to the strongest baselines (HAC, Compact3DGS). The hybrid SFP combining 3D hash grids and 2D planes is lightweight yet expressive, effectively capturing multi-scale spatial correlations. Implicit representation makes the information more compact.

**Weaknesses:**

Although ablations are provided, a deeper investigation into trade-offs between SFP resolution, residual dimension, and compression ratio would strengthen the analysis.

The paper lacks a comparison with state-of-the-art methods. The authors’ explanation for not comparing with HAC++ is insufficient, and the current experimental baselines appear selectively chosen. Moreover, the so-called predictive techniques that encode features into spatial factorizations are not introduced for the first time in this work. Methods such as TensorRF and CCNeRF have explicitly shown that spatial factorization of implicit features can reduce data requirements. I agree that adding residual features to anchors may be a highlight of this method, but I do not think this contribution alone is sufficient to justify the lack of comparison or discussion with methods like HAC++ and HEMGS.

**Questions:**

What is the specific storage capacity ratio between the SFP and the residual features after compression?

---

### Official Review · Reviewer_SBy2 · 2025-11-01

**Soundness:** 3
**Presentation:** 3
**Contribution:** 2
**Rating:** 4
**Confidence:** 4

**Summary:**

This paper introduces Predict-GS, a novel compression framework designed to significantly reduce the storage size of 3D Gaussian Splatting (3DGS) models. The core problem Predict-GS tackles is spatial redundancy: in existing "anchor-based" 3DGS methods, nearby anchors in a 3D scene have highly correlated features, but each feature is stored independently. Predict-GS solves this by introducing Prediction with a Spatial Feature Pool (SFP) and Residual Compensation, instead of encoding the entire feature for every anchor, the method only needs to store the compact SFP and the small residual values, drastically reducing the bitrate. The paper also introduces a prediction-tailored progressive training strategy, where training starts with downsampled images to help the SFP learn the global scene structure before gradually introducing high-resolution details and the residual compensation module. On standard benchmarks, Predict-GS achieves a 58x compression over the original 3DGS and a 16.91% bitrate reduction over the previous state-of-the-art compression method (HAC), while maintaining or even slightly improving rendering quality.

**Strengths:**

- The method integrates compression into the training process rather than applying it post-training, effectively leveraging the benefits of joint optimization.
- The approach achieves a high compression ratio without significant performance degradation, and in some metrics even surpasses the baselines.

**Weaknesses:**

- While 3DGS can achieve hundreds of FPS, the compressed model only reaches roughly half of that. This contradicts the typical expectation that compression should improve speed. Additionally, 3DGS still delivers better performance in some scenarios, which raises questions about the practical benefits of the proposed compression.
- The paper’s technical description is somewhat unclear. For example, the origin of the residual feature is not fully explained. Figure 2 suggests it comes from the anchor, but the anchor feature appears to be derived from the Spatial Feature Pool. This relationship and data flow should be clarified, as the current explanation is confusing.
- The proposed two-stage progressive training strategy with cosine annealing, while effective, increases the complexity of the training pipeline compared to standard 3DGS training. The paper briefly mentions that the method is “even 4% faster than HAC,” but it would be important to provide training time comparisons against 3DGS and ScaffoldGS as well, given the added pipeline complexity.

**Questions:**

I find that the paper does not clearly explain an essential component of the method—specifically, how the residual feature is obtained. This part is central to the approach, and the current description leaves ambiguity regarding the feature extraction process.

Additionally, since this work focuses on compression, inference speed should be a key metric. However, the proposed model runs slower than vanilla 3DGS and ScaffoldGS, which undermines the motivation for compression. The training time is also not reported, despite the added pipeline complexity. The authors should address and justify these weaknesses more thoroughly.

---

### Official Review · Reviewer_MpJQ · 2025-11-02

**Soundness:** 3
**Presentation:** 3
**Contribution:** 3
**Rating:** 6
**Confidence:** 5

**Summary:**

This paper proposes Predict-GS, a method that compresses Gaussian scenes into just a few tenths of MB. To this end, Predict-GS first encodes the scene into a structurally compact representation: Gaussian anchors, which is inspired by Scaffold-GS. To further compress anchor features, Predict-GS introduces a Spatial Feature Pool (SFP) consisting of multi-resolution 3D hash grid for storing coarse structure information and high-resolution tri-planes for capturing fine-grained, high-frequency details. Although the SFP effectively compresses feature information, retrieving features from the continuous feature volume and plane leads to some loss of detail. To recover anchor-specific, localized high-frequency information, Predict-GS proposes a Multi-Head Gaussian Residual Decoder (MGRD), which refines Gaussian-level attributes of shape (MGRD_s) and appearance (MGRD_a) respectively. To further stabilize the training of SFP and MGRD, Predict-GS adopts a staged training strategy, where first learning coarse scene structure by optimizing SFP alone, then fine-grained details by optimizing MGRD, and a progressive downsampling schedule for training signals. This schedule begins with downsampled images at initial step and gradually restores them to full resolution following a cosine annealing schedule. Through extensive evaluations on multiple benchmark datasets, Predict-GS demonstrates high reconstruction quality while requiring minimal storage and shows necessity of proposed SFP, MGRD, progressive training method via ablation study.

**Strengths:**

This paper presents a novel idea that combines structural 3DGS compression with feature encoding, demonstrating impressive bitrate savings. Overall, the text is well written and easy for readers to follow, and the experiments effectively validate the proposed approach. Moreover, the authors provide thoughtful discussions on limitations, the trade-off between computational efficiency and compression, implementation details, supporting experiments for hyperparameter selection (e.g., initial downsampling scale), and clear distinctions from related concurrent works such as HEMGS and HAC++. These aspects together justify the authors’ proposal of Predict-GS.

**Weaknesses:**

I think this paper is well written, but to enhance its soundness, please address my concerns listed in the Questions section.

**Questions:**

* Many readers may wonder about the computational latency of this method. I suggest including a comparison of training duration and rendering speed in Table 1.
* The authors discuss four primary strategies for 3DGS compression: compact structure, Gaussian pruning, quantization, and entropy coding. I believe that **controlled densification of Gaussians** is also a major direction for 3DGS compression, as its goal is to achieve a smaller number of Gaussians with highest reconstruction quality. Therefore, I suggest the authors additionally discuss recent research on this direction in the main paper and include papers in this direction in Table 1, such as:
  1. Mallick et al., Taming 3DGS: High-Quality Radiance Fields with Limited Resources, SIGGRAPH Asia 2024
  2. Chen et al., DashGaussian: Optimizing 3D Gaussian Splatting in 200 Seconds, CVPR 2025
* Furthermore, I have a specific question: compared with density control methods, can it be said that SFP-based compression performs better (in both reconstruction quality and storage efficency)? Or are these methods orthogonal and potentially complementary when applied together?
* I am also curious about how many Gaussians (anchors) were propagated to achieve the reconstruction quality reported in Table 1. It would be valuable to compare the number of Gaussians to clarify where the storage size reduction originates.
* In Table 1, the selected baselines are somewhat outdated. I suggest including the following additional, more recent baselines:
  1. Chen et al., HAC++: Towards 100× Compression of 3D Gaussian Splatting, arXiv
  2. Wang et al., ContextGS: Compact 3D Gaussian Splatting with Anchor-Level Context Model, NeurIPS 2024
  3. Chen et al., MEGS2: Memory-Efficient Gaussian Splatting via Spherical Gaussians and Unified Pruning, arXiv

---

### Official Review · Reviewer_TUEV · 2025-11-03

**Soundness:** 3
**Presentation:** 3
**Contribution:** 1
**Rating:** 2
**Confidence:** 5

**Summary:**

This work introduces a rate-distortion-optimized 3DGS compression framework in order to save transmission bandwidth and/or storage space. Like many prior works in this area, it is built on Scaffold-GS. Additionally, it introduces multi-scale 3D dense grids and 2D plans to perform a prediction of anchor attributes. The resulting residues (attached to anchors) are further transformed non-linearly by an MLP into their latent representations for entropy coding

**Strengths:**

The progressive training strategy is interesting and relatively novel

**Weaknesses:**

Many SOTA methods, such as CAT-3DGS, CAT-3DGS Pro, and ContextGS, are missing.

The idea of using hash tables to query anchor features (e.g. Eqs. (1) and (2)) has been explored in HAC and its sequel. The proposed method is more like the direct substitute approach mentioned in the HAC paper. That is, SFP predicts directly features during reconstruction.

The idea of using residual coding is not new either. It is to be noted that many prior works in this area model the probability density functions of anchor attributes by Gaussian.  The predicted means are often subtracted from the corresponding anchor attributes before the resulting residues are further entropy coded by a zero-mean Gaussian. In this sense, they already adopted the notion of residual coding. The major distinction between this work and these prior works consists in the use of MGRD’s to decode residual signals in a non-linear fashion.

Line 220: The statement that anchor coordinates with similar spatial features may be mapped to the same entry appears problematic. It depends on how the hash function is designed.

**Questions:**

How does the proposed method compare with the more recent publications such as CAT-3DGS, CAT-3DGS Pro and ContextGS?

---

### Meta-Review · Area_Chair_psfC · 2026-01-04

**Summary:**

The majority of the reviewers suggest rejection, and there is no rebuttal.

**Reviewer Scores:**

No change.

---

### Decision · Program_Chairs · 2026-01-26

Reject